# Developing Preservice Chemistry Teachers' Engagement with Sustainability Education through an Online Project-Based Learning Summer Course Program

**Maria Paristiowati** [1,*], **Yuli Rahmawati** [1], **Ella Fitriani** [1], **Justinus A. Satrio** [2]
**and Nur Azizah Putri Hasibuan** [3,*]

1    Chemistry Education Department, Universitas Negeri Jakarta, Jakarta 13220, Indonesia;
     yrahmawati@unj.ac.id (Y.R.); ella.fitriani@unj.ac.id (E.F.)
2    Chemical Engineering Department, Villanova University, Philadelpia, PA 19085, USA;
     justinus.satrio@villanova.edu
3    Khalifa IMS Secondary School, South Tangerang 15228, Indonesia
*    Correspondence: maria.paristiowati@unj.ac.id (M.P.); nurazizahputri@khalifaims.sch.id (N.A.P.H.)

**Abstract:** The aim of this research was to develop the sustainability competencies of preservice chemistry teachers' through the use of a project-based learning model. Preservice chemistry teachers were engaged in a summer course program in collaboration with national and international universities. The summer course program was conducted online due to the COVID-19 pandemic. The research involved 26 preservice chemistry teachers from a pedagogical university in Jakarta, Indonesia, which joined with other university students from other universities in Indonesia, America, Thailand, and Malaysia. We used a qualitative methodology. Data were collected through interviews, questionnaires, observations, preservice chemistry teachers' portfolios, and reflective journals. The data were coded into themes and interpreted to reveal that all students engaged successfully in developing their sustainability perspectives, environmental awareness, project development engagement, communication, and collaboration skills. Meanwhile, the preservice chemistry teachers engaged in developing their project in an online summer course program within the framework of sustainability.

**Keywords:** sustainability; project-based learning; preservice chemistry teacher

## 1. Introduction

In the current era of globalization, there is a heightened demand for student teachers to demonstrate 21st-century skills and environmental insights to foster a sense of responsibility towards the environment. Education for Sustainable Development (ESD) is a toolkit, developed by UNESCO, that can be used to increase environmental awareness. The ESD toolkit uses three pillars [1] to guide students to take responsibility and make decisions related to environmental, economic, and social problems [2]. Sustainable development is described as development that meets the needs of the present without compromising the future generation's ability to meet their own needs. ESD means creating and living a human life on Earth in a way that does not damage life, but that preserves its various life forms for the future—not only for future human life [3]. The purpose of ESD is to promote education as a crucial tool in preparing today's generation to become responsible citizens so that future generations can continue to shape a sustainable society.

The principle of sustainable development focuses at the same time on human livings today and in the future. Sustainable development is both a world view and a method of solving worldwide problems that emphasizes three dimensions: economic, social, and environmental [4], and distinguishes two approaches: analytical and ethical [3]. Sachs said that "Sustainability development is both a way of looking at the world, with a focus on the interlinkages of economic, social, and environmental change, and a way of describing our

shared aspiration for a decent life, combining economic development, social inclusion, and environmental sustainability. It is in short both an analytical theory and a 'normative' or ethical framework" [4]. The evaluation of the impact of the Decade of ESD, that ESD was increased but a few states could report full implementation across education systems, as well as across policies and planning. However, the educational effort has not been radical enough to address the most urgent problem of time [5–7]. There is obviously still much to accomplish and many challenges to face [8] to improve ESD implementation.

All levels and domains involved in education, including chemistry education, can contribute to ESD [9]. Chemistry education can play an important role in promoting sustainable development. A practice known as green chemistry provides guidance in applying more sustainable chemistry practices to environmental issues [10]. Many areas of business have chosen to implement a green chemistry approach even though they are not compelled to. However, education is needed to build a society that is knowledgeable and concerned about negative media reports relating to social–science issues [11]. Future chemistry teachers must become more skilled at actively participating in public debates about chemistry and technology. They have a role to play in formulating social decisions towards a sustainable future and in changing societies' attitudes and behaviors towards a sustainable lifestyle.

The implementation of ESD-based pedagogy in schools remains limited in the face of more content and practice-based chemistry education designed to meet the demands of competence in a content-intensive curriculum. It is not enough to simply overlay environmental and basic chemistry issues related to sustainability onto the content and context of current chemistry curricula. ESD requires the application of a skills-oriented teaching paradigm to promote ESD practices, which is more than just education about sustainable development [12]. Therefore, a socio-scientific-based curriculum, specifically focused on sustainability issues, needs to be developed [13]. Chemistry education does not always describe the social and economic dimensions of sustainable development sufficiently, so making changes to the orientation of ESD is necessary [14,15].

Chemistry education should feature prominently in ESD due to the role played by chemistry and the chemical industry in everyday life [16,17]. Chemistry education students are expected to develop an understanding of the role of chemistry in society and be able to evaluate how chemistry can contribute to community sustainability and support the management of natural resources [9,18]. For decades, teachers have been cited as key agents in the sustainability process [19]. Analysts have asked for a stronger emphasis on teacher education to prepare them for this important task. Yet, most teachers still lack this respect [20–24].

Teachers' and students' knowledge of ESD was vaguely informed in the theoretical sense, and only a few of them possessed any clear theory-supported concept of either sustainability or ESD [25]. Education settings need to become more proactive in sustainability, and to make that happen, ESD needs to be a *core concept* in teacher education, and its implementation must be obvious in policy, campus practice, research, teaching, and learning [3], especially for preservice chemistry teachers. According to [26], teachers are the most important factor in education reform. What teachers think, believe, and know will affect their teaching. For this reason, the development of teacher competencies is an essential key to successfully growing the value of ESD in learning.

Knowledge about ESD is an essential element of 21st-century learning; therefore, it is crucial that preservice teachers understand how to implement it in schools throughout the learning process. The contribution of chemistry education to support ESD includes developing prospective chemistry teachers' understanding and their ability to apply appropriate pedagogies. Personal, social, and professional competencies are required for a paradigm shift, where knowledge about ESD is integrated and reflected in the preservice teachers' approach to chemistry education in a way that instills the value of sustainability into their students as early as possible. Burmeister at [16] present four strategies for implementing sustainable development issues in the field of chemical education by (1) applying green

chemistry principles to practicum, (2) adding sustainability strategies as content in chemical education, (3) including socio-scientific issues and controversies in teaching, and (4) using chemical education as part of ESD-driven school development.

Successful 21st century learning demands competencies in critical thinking, collaboration, communication, and creativity. In order to achieve these competencies, students must be provided with a supportive process that values student-centered, active, and collaborative learning. Project-based learning (PjBL) is one example of a student-centered model where students learn to build their own learning experiences independently [27]. Learning is focused on solving challenging problems through a series of complex tasks which involve students investigating essential ideas, designing, solving problems, and making decisions by working independently and collaboratively to create realistic products or presentations. The PjBL process involves students using creative and critical thinking skills, developing opinions, and drawing conclusions [28].

In this study, 26 preservice chemistry teachers from a pedagogical university in Jakarta were engaged in a summer course to improve their understanding of sustainability project-based learning. The summer course involved students from other universities in Indonesia, America, Thailand, and Malaysia. Students worked on projects oriented to issues of sustainable development. The study was expected to develop preservice chemistry teachers' competencies through a sustainable-development-project-based learning model. The study created a competency development model for chemistry teacher candidates through project-based learning oriented towards sustainable development.

## 2. Materials and Methods

### 2.1. Research Design

A qualitative research method, adopted from [29], which defines a case study as "an in-depth exploration of a bound system such as an activity, event, process or individual", was used in the study. An interpretivist paradigm was used to produce descriptions and explanations about the participants' engagement throughout the program. Multiple data-collection strategies, such as observations during learning activities, interviews, reflective journals, documenting assigned tasks, video, photos, and observation sheets, were employed to explore the participants' engagement.

The participants in this research were preservice teachers who studied in a pedagogical university in Jakarta. A non-probability sampling strategy or criterion-sampling method was applied to measure their willingness to participate in this research. The study group consisted of 26 female undergraduate and master chemistry education students. No male students participated in the research project, which supports the literature that attracting men into the teaching profession is a real challenge. The subjects participated voluntarily in this study, and their identity was kept confidential. Descriptive information about the study group is presented in Table 1.

**Table 1.** Demographic characteristics of participans.

| Characteristic | Parameter | Frequency (N) | Percentage (%) |
|---|---|---|---|
| Gender | Female | 26 | 100% |
|  | Male | 0 |  |
| Years of Studies | 2017 | 11 | 42.3% |
|  | 2018 | 15 | 57.7% |
| Cycle of studies | Bachelor's Degree | 25 | 96% |
|  | Master's degree | 1 | 4% |

The research was conducted online due to COVID-19 restrictions imposed in December 2019. The lessons and materials were delivered through a range of online platforms, such as ZOOM, WhatsApp, Google Classroom, and YouTube. The summer course program ran from 5 October 2020, until 14 November 2020. The research was conducted in three

stages: a preliminary stage, a research implementation stage, and a final stage. The research flow is shown in Figure 1.

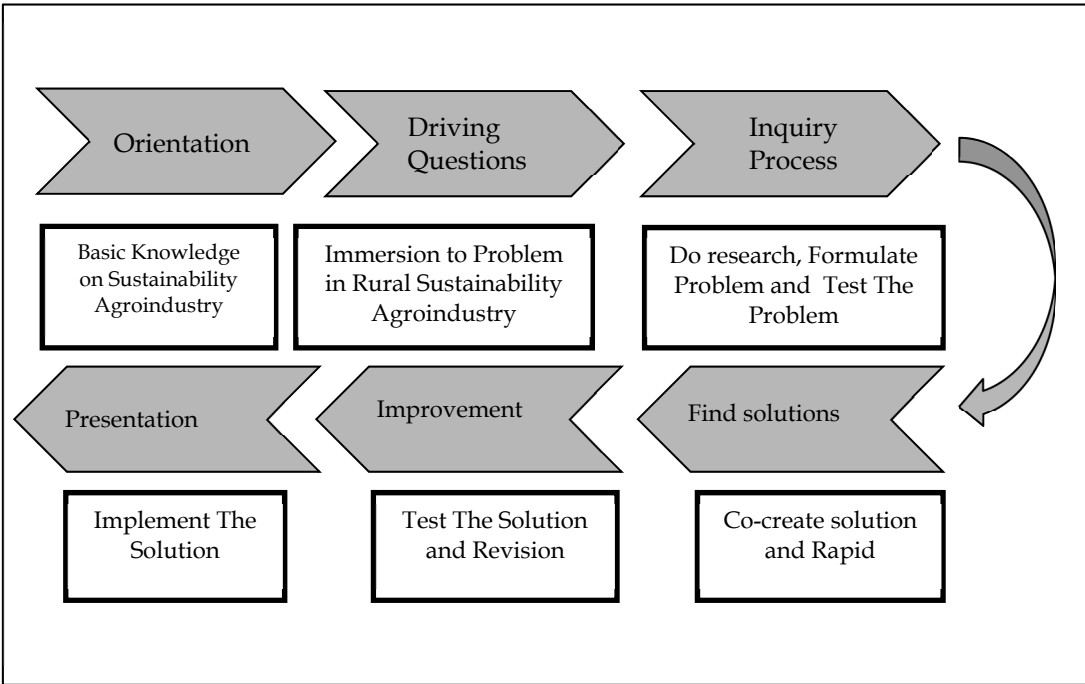

**Figure 1.** Summer course program outline and learning activities [30].

At the preliminary stage, two types of questionnaires were used to determine the participants' prior knowledge of ESD concepts and Project-Based Learning (PjBL). The course program followed an inquiry process where a driving question challenged students to find solutions, create a product with sustainable value, receive feedback from the teacher, and use it to revise their solution to present at the end of the course. The learning process began with a presentation of sustainable development material in general before focusing on a more detailed explanation of sustainable development in the agro-industry. Input from practitioners or entrepreneurs in the coffee, chocolate, and tea industry provided students with information regarding the sustainability of the agriculture industry (agro-industry). Virtual visits to coffee plantations provided the participants with a realistic feel for the atmosphere and conditions that occur in the agro-industry sector. The series of activities at this stage of the process enabled the participants to develop projects using agro-industry products that integrate with sustainable agro-industry business ideas. The participants worked collaboratively to develop the project over a two-week period. They elaborated on one or more lessons in detail, prepared relevant resources and materials, and carried out key parts of the project assignment to serve as a model for their peers. The activity was presented to the whole group and was followed by a discussion in a 2x2 hourly session. Both presentations were peer- and supervisor-rated.

Based on the first and second stages, the process was summarized during the final phase into a project plan to be included in a competition called the Festival of Agroindustry. The project plans were also presented at the end of the program to show appreciation for the best projects selected by the Summer Course Program Assessment Team. At the end of the program, an evaluation form was provided to all participants regarding the process, development, and results obtained by them during the online summer course program. Semi-structured interviews were conducted to determine participants' views regarding the program in more detail by providing several open-ended questions and additional questions to obtain in-depth answers. Reflective journals using guiding questions were also completed by the participants.

*2.2. Online Summer Course Program*

Online Summer Course program activities were held by a university in Indonesia collaborating with several national and international universities from America, Indonesia, Thailand, and Malaysia. The total participants were 75 students with 27 preservice chemistry teachers who joined the research. This program offered online classes and virtual field trips with the theme of sustainable development. The lecturers came from numerous universities and experts from the Indonesian enterpreneur and Research Institute.

The program was designed to provide students with the knowledge and ability to design a sustainable rural agro-industry chemistry project. Students would learn about the development of sustainable agro-industry products and processes, which add value to local commodities and support the rural agro-industry. The program teaches students sustainability concepts, engineering, and the business aspects of a sustainable agro-industry.

The research program was initially planned for delivery in a face-to-face learning environment; however, because of the pandemic, the intended participants, lecturers, and other stakeholders were unable to travel to Indonesia. The materials, assignments, and activities were, therefore, successfully adapted for delivery and administration online.

The program was designed to provide students with knowledge of sustainable development by integrating chemistry, industry, environment, and management processes. Students learned from hands-on experience and practice about product development and processes aimed at adding value to an environmentally friendly product, system, and management. Students were challenged to use design thinking and to apply it to the field of chemistry. The research team assessed the preservice teachers' competencies during and after the program.

*2.3. Data Collection and Analysis*

Data were obtained using a questionnaire, an observations sheet, an interview, a reflective journal, and an evaluation questionnaire.

2.3.1. Questionnaire

Questionnaires are useful for measuring observed natural or social phenomena. The natural or social phenomenon in question is a variable in a study. The questionnaire used in this research was self-administered online at the beginning of the study and before providing the program material to obtain the participants' perceptions and interest in learning ESD-based PjBL. The instrument was compiled based on the PjBL and ESD grids, each with its own criteria for ESD understanding, ESD and chemistry relations, PjBL in teaching chemistry, and PjBL understanding. The instrument used a Likert scale with "Very agree" to "Not agree" statements with a scale from 1 to 5. The questionnaire results were analyzed by counting the value of the questionnaire and distributing the value into "low", "middle", or "high" levels of understanding and interest in ESD and PjBL.

2.3.2. Semi-Structured Interview

To complete data collection, a semi-structured interview was used to elicit students' ideas and views on ESD, the online summer course, project development, and a multicultural environment. All interviews were recorded, and the results were fully transcribed by the researcher with a focus on the participants' engagement through the program. The following are examples of questions posed in the interview about ESD, online summer course, project development, and multicultural environment: "What do you think about ESD?"; "What impressions did you feel in the project planning process?"; "How about working, discussing, and collaborating with people you just met?"

2.3.3. Reflective Journal

A reflective journal was introduced in the middle of the program and at the end to capture the participants' reflections on the challenges they faced and their level of engagement during the learning process. One example of a prompt in the reflective

journal was "write down your reflection in order to develop your understanding about sustainability relation with chemistry after joining this program".

### 2.3.4. Observations

Observations are useful for gaining a deeper understanding of the learning process and the participants' achievements. Observations were conducted throughout the summer course program with supporting documentation obtained from interviews, video presentations, photos, a themed schedule, and the participants' task assignments.

A qualitative data-analysis technique, including data reduction, data display, and conclusion drawing/verification, was used in this study [31]. The participants' engagement during the program implementation was analyzed from data collected by observations, reflective journals, interviews, and task assignment portfolios.

The findings, based on the research questions, were displayed and verified for validity and trustworthiness by using credibility techniques such as prolonged engagement, persistent observation, progressive subjectivity, and member checking [32].

### 3. Results and Discussion

This section reports the project-based learning descriptions, prior knowledge of students upon the project, perceptions on the online summer course, and impacts on preservice chemistry teachers. The impacts of preservice chemistry teachers are analyses based on thematic analysis of data from data collections in response to the research question: Can a Preservice Chemistry Teachers' Engagement with Sustainability Education improve Through an Online, Project-Based Learning Summer Course Program?

Thematic analysis of the impacts of preservice chemistry teachers indicates that there are four aspects, which are sustainability perspectives, environmental awareness, project development engagement, and communication and collaboration.

### 3.1. Project-Based Learning Descriptions

PjBL learning was applied in stages throughout the research project guided by experts on how to use a structured thinking pattern to come up with an idea to develop into a project or business plan, how to manage an idea, and create an innovative solution. An important part of this research focused on how prospective chemistry teachers could develop sustainable projects to develop students' skills. The project development or "mini project" created new approaches to agro-industry products based on sustainability. During the initial stage of developing an idea, the participants were asked to prepare a project plan (project charter) using design thinking to serve as a guide for developing their project.

During the following stage, the participants were asked to review their project charter to include desirable, feasible, and viable aspects of sustainability [33]. One of the project groups reported that their project was desirable because the project would reduce waste and generate alternative income for farmers. They claimed their project was feasible because of the continuous availability of raw materials and the antioxidant content in organic compounds. They considered that the product would benefit the health of users. This step confirmed how confident the students were about their project proposal (see Figure 2).

The participants developed many project plans through the program, including one called COHUMA (*Coffee Hush Mask*). The group created an innovative face mask made from coffee husks to reduce acne-causing bacteria, disguise black spots, and provide an anti-aging effect on the face. In addition to using coffee husks as the main ingredient and to maximize the performance of the mask, the participants added vitamin E to moisturize, soften, and brighten the facial skin and to fade acne scars. The product design can be seen in Figure 3 and the instructions for use can be seen in Figure 4.

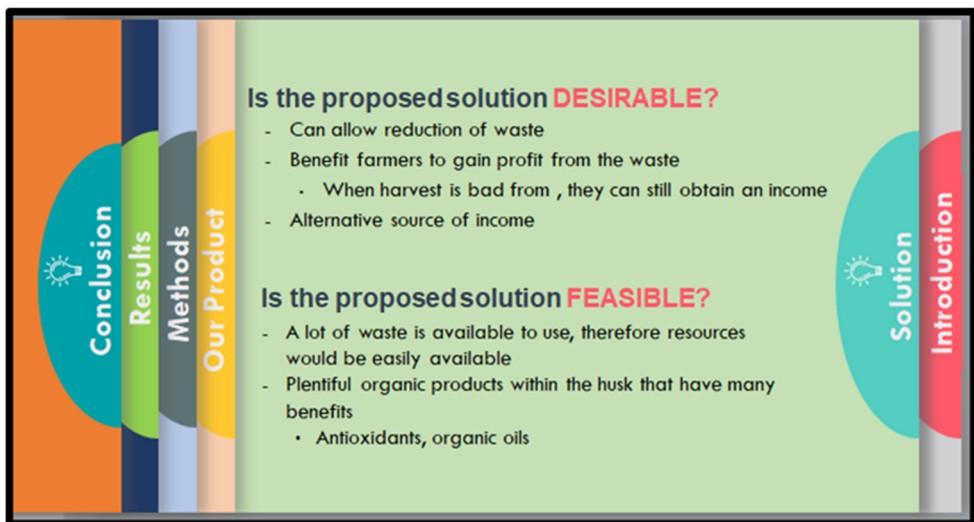

**Figure 2.** Participants' slide documentation, Green Tea Group, 15 November 2020.

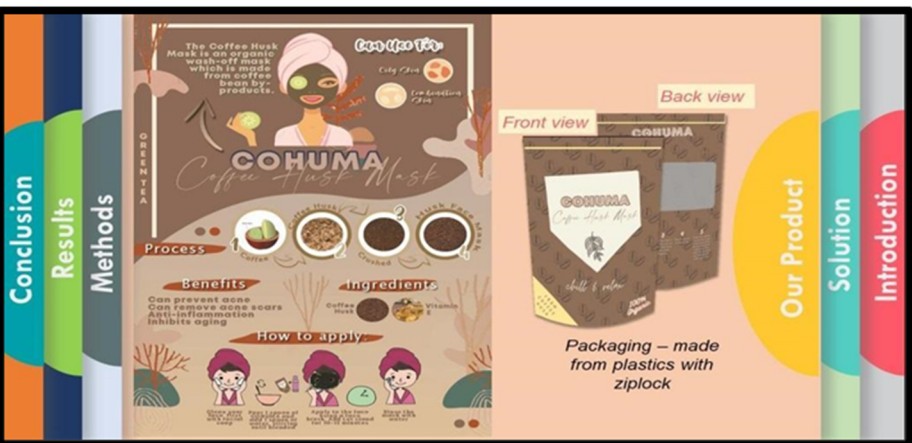

**Figure 3.** Product design.

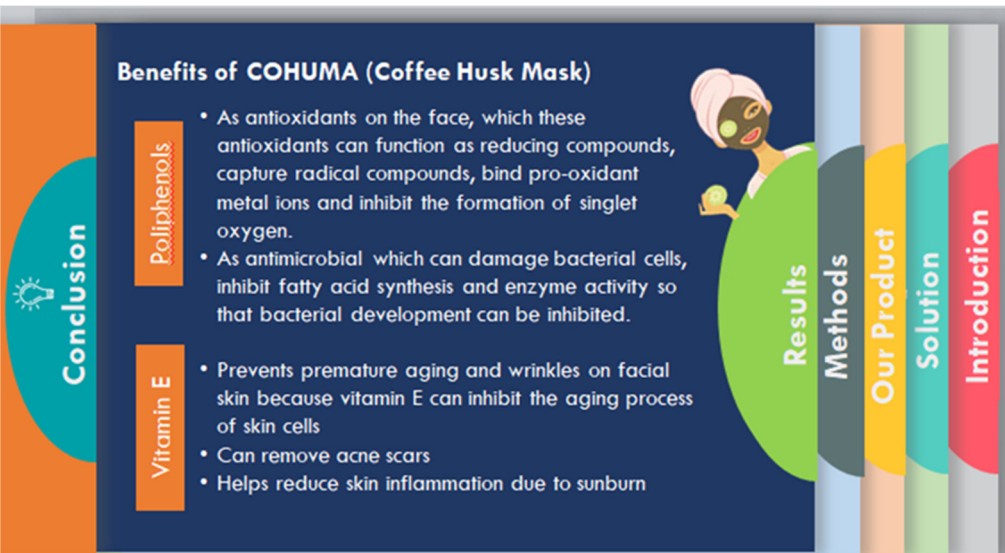

**Figure 4.** The benefits of using the mask.

As shown in Figure 4, the chemical concept that students explored in this project was the antioxidant value of coffee, which has the effect of reducing compounds, scavenging

radical compounds, binding prooxidant metal ions, and inhibiting the formation of singlet oxygen. The project demonstrates the integration of chemical concepts with economic potential that contributes to reducing environmental impacts.

The picture below (see Figure 5) is one of the presentation documents the participant group provided to show their project development at the end of the summer course program.

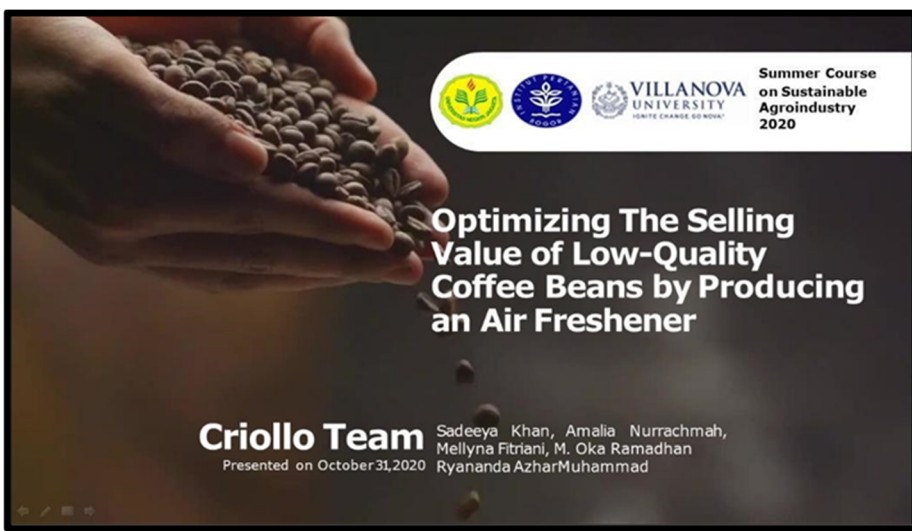

**Figure 5.** Project development by preservice chemistry teachers [30].

After the program had run for two weeks, the participants were asked to complete a reflective journal. Reflective journals are a useful student-centered activity that provides opportunities for self-education. Journals can provide inspiration and motivation to students who use them purposefully to reflect on past experiences that were successful or meaningful for them, thus increasing their desire to continue learning [34].

The reflective journals were analyzed by coding the participants' responses to sustainability, project-based learning models, multicultural-environment-based learning, and online-based learning. The concept of ESD is in line with the idea of chemical education by Holbrook and Rannikmae in [2], which includes a shift from merely learning chemistry knowledge to developing skills.

The ESD process encouraged students to develop high-level cognitive skills through decision making, solving problems, taking responsibility, and evaluating their thinking during chemistry lessons. As shown in Figure 3, participants reported that their knowledge and awareness of sustainability improved, as did their problem solving, critical and creative thinking, communication, and high-order thinking skills (HoTS).

Various activities of the online summer course program were documented, as shown in Figures 6 and 7 below.

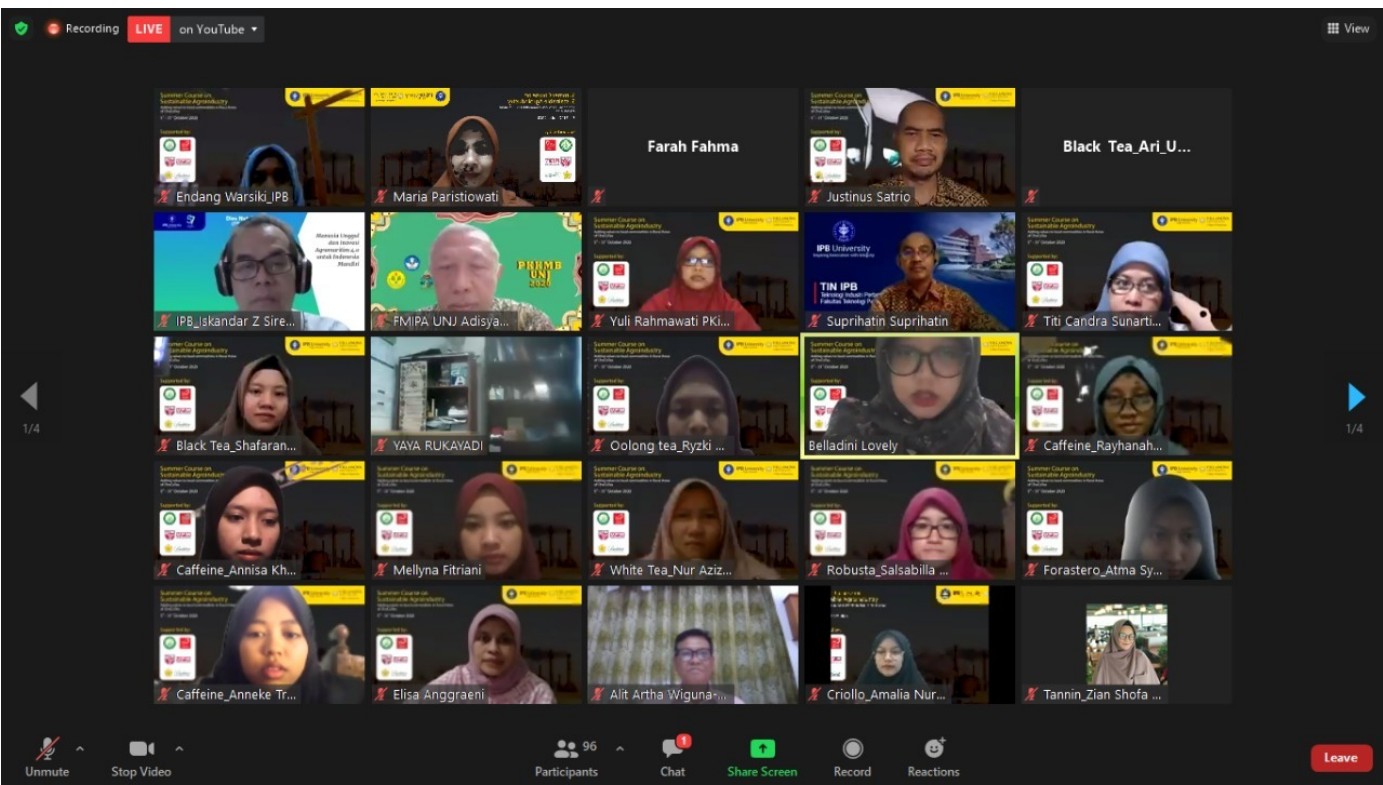

**Figure 6.** Summer course 2020, opening ceremony with all participants and lecturer.

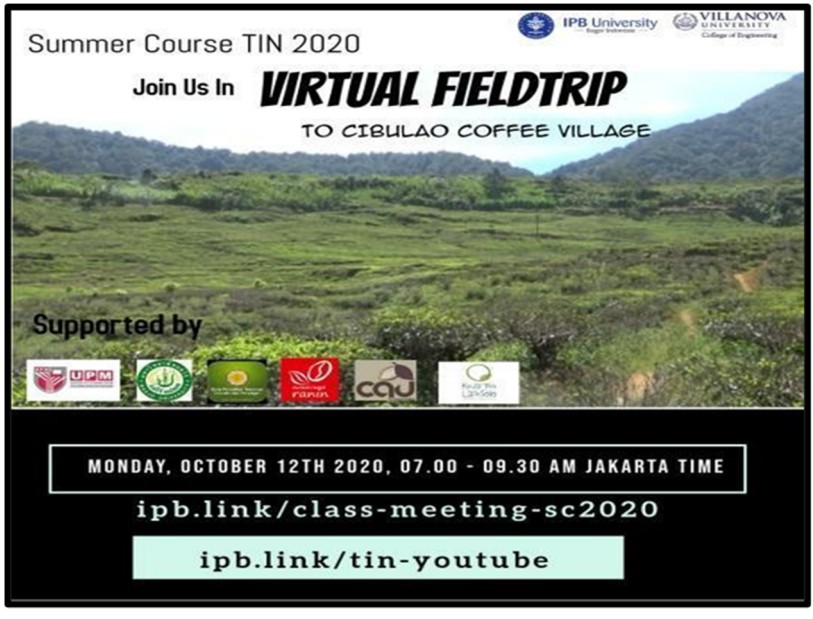

**Figure 7.** Summer course activity, virtual field trip to cibulao coffee.

### 3.2. Prior Knowledge of ESD and PjBL

Before starting the summer course program, a more detailed picture of the participants' knowledge of chemistry, ESD, and PjBL, and how interested they were in participating in the competency development program, was created through the use of a questionnaire. A second questionnaire applied at the end of the program revealed whether the participants were fully engaged in the program or not.

The first questionnaire asked about (1) knowledge about ESD; (2) ESD in life; (3) relevance of ESD with chemistry; (4) relevance of ESD with chemistry education; (5) hope

after learning ESD; (6) interest in learning ESD. The researchers considered these points important to determine preservice teachers' engagement in this program.

According to data analysis obtained from the first questionnaire, the participants' knowledge about ESD fell at the middle level (73.96%), while their interest in learning ESD was high (82.29%) (see Figure 8). The graph below shows that knowledge and integrated ESD in life scores at the middle level, while ESD relevance with chemistry, the participants hope in ESD, and ESD relevance with chemistry education score high, indicating that the participants were highly motivated to learn ESD at the beginning of the program.

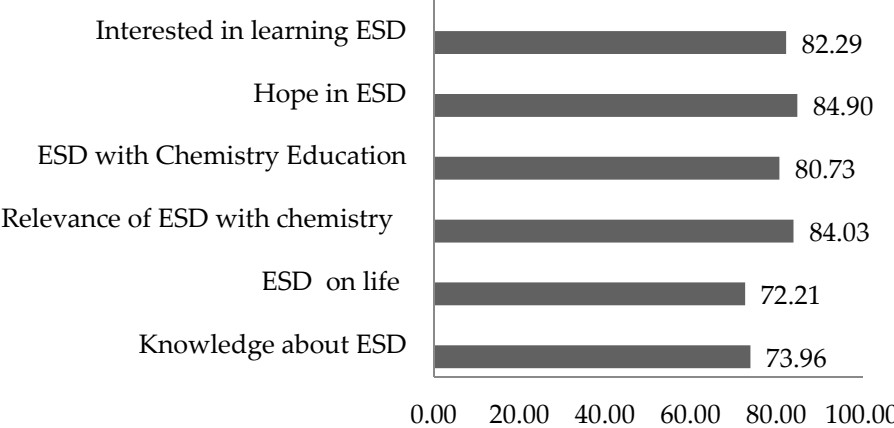

**Figure 8.** Participants' response toward ESD [30].

In-depth interviews revealed that students' knowledge of sustainable development (SD) prior to participating in ESD-based PjBL learning was only related to the chemistry field. One of the participants said that:

*"Before, I was not very aware of sustainability, especially in the field of agro-industry. This program increased my knowledge about SD in the agro-industry field. I never thought that sustainability was also carried out in other fields besides chemistry."* (Participants 1 (P1), Interview 20 November 2020)

The interviews revealed that students did not know that the agro-industry can also contribute to developing SD values or that SD is a multidisciplinary field in which every field of science can be developed based on the socioeconomic–environmental principles of SD and other complex, interrelated aspects [3]. Sleurs at [35] suggests that " if the teacher has the intention to take the issue of sustainable development seriously they will also link the issue to the economic, social, cultural and political aspects", then sustainable education in the field of chemistry, for example, must be interrelated with other disciplines if sustainable values are to be developed.

According to the questionnaire data, the participants considered ESD relevant if integrated into the chemistry curriculum. The motivation for chemistry teacher candidates to join the program scored a high of 84.03%, as indicated by the following interview transcript.

*"I think as a prospective chemistry teacher it is important to study SD because studying chemistry has something to do with the environment, so the students can use chemical concepts to solve problems in the surrounding environment"* (P3, interviewed 22 November 2020)

The participants could see that SD is necessary and useful to study and apply in chemistry learning, which motivated them to join the program even though the material was not fully relevant to the chemistry field. Through project-based learning, ESD can be included as a discussion theme, or students can work on projects oriented to the principles of ESD. Research shows that a project-based learning model can improve students' science process skills [36–38]. Participants' responses on knowledge of PjBL can be seen in Figure 9 below.

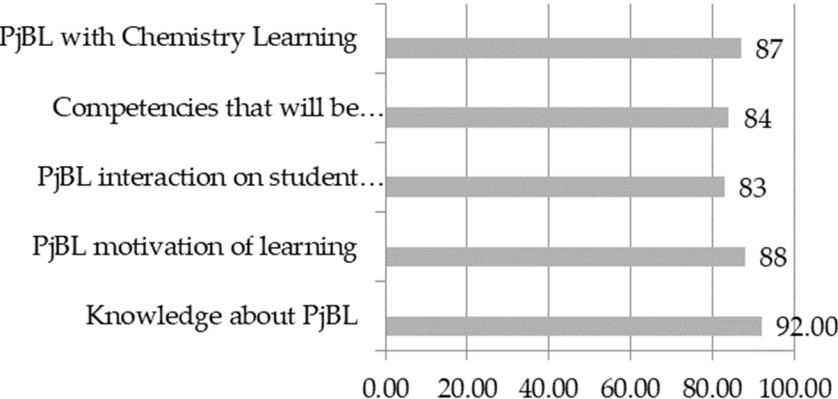

**Figure 9.** Participants' response through PjBL in chemistry learning [30].

The results of the PjBL questionnaire (see Figure 3) indicate that the participants were knowledgeable about PjBL (92%), PjBL motivated learning, participants understood the relationship to chemistry, and PjBL encouraged students to learn actively. On average, the preservice chemistry teachers were very familiar with the PjBL model. It has positive impacts as project-based learning develops students' critical-thinking skills more effectively than conventional learning models [39], develops students' 21st-century skills, and develops teachers' professional abilities in learning and assessing 21st-century skills [40]. Research conducted by Devkota et. al., [41] claims that implementing project-based learning can increase learning motivation and student engagement and change students' attitudes from passive to active learners.

### 3.3. Online Summer Course

The use of online learning systems has become necessary because of the COVID-19 pandemic dominating headlines since December 2019. School closures in over 180 countries have laid bare inequalities in education, deficiencies in remote learning, the cost of a digital divide, and have revealed the important role schools play in students' health and well-being [42]. UN Secretary-General Antonio Guterres has called upon governments to "build back better" after the current crisis by creating more sustainable, resilient, and inclusive societies. This must include education, as societies cannot transform if what we learn and how we learn it maintains the status quo.

ESD empowers learners of all ages to change their thinking and to work toward a sustainable future by addressing environmental integrity and economic viability and demonstrating how to work towards a more just society. ESD requires a reassessment of how education can form critically minded, empathetic students who can work together to solve problems and act on present and future local and global emergencies [42].

While studying online was not new for the students, they were, nevertheless, confronted by problems related to poor connectivity that made understanding the material presented during online classes difficult. Despite the challenges, students reported feeling enthusiastic about joining the class because they gained new insights, developed collaborative skills, and increased the scope of friendships, as demonstrated by a student's response below.

> *"It's interesting because you can be in groups even through ZOOM apps, do assignments at the GCR, have group discussions with the Whatsapp group, and maybe the network that has the most problems with climate or weather changes in Indonesia can determine a provider network. The quota that sometimes runs out is also a problem in implementing this online class. Overall, it is fun but sometimes a little boring if we have just paid attention for three hours. Maybe offline is more fun."* (P7, Interviewed 22 November 2020.)

The online summer course brought opportunities to the class that were different from those gained by face-to-face teaching. With the pandemic still underway, governments around the world must continue to reassess learning systems to improve education alongside the economy and to fight environmental problems with strategies such as ESD.

ESD can prepare students to learn effectively in complex situations to promote sustainable development. By using ESD as a roadmap for future education, teachers can strengthen their students' capabilities to tackle multiple challenges. The commitment to ESD must include learning for all learners in all contexts, regardless of gender, location, socioeconomic status, or access to technology. The online summer course provided participants with an opportunity to engage deeply with ESD principles at a time when the world was in crisis.

*3.4. Impact on Preservice Chemistry Teachers*

The impact on preservice chemistry teachers is found within four themes of sustainability perspectives, environmental awareness, project development engagement, and communication and collaboration, as explored below.

3.4.1. Sustainability Perspectives

ESD-based learning is expected to develop student competencies in knowledge, behavior, and attitude, which are collectively referred to as sustainable competence [43]. Sustainable competence relates to cognitive components such as knowledge and understanding related to environmental, social, economic, political systems, and higher-order thinking skills such as reasoning and synthesis, as well as social abilities, values, and emotions, also known as the affective domain.

The preservice chemistry teachers' competencies that emerged during and after the summer course program described values, beliefs, and judgements, as recorded in the excerpts below:

The summer course program identified that the participants' changed their mindset from being unconscious about the importance of sustainability to becoming more aware that applying sustainability to other fields is essential. This finding is shown by the participants concerned about the sustainability value of a product, as confirmed by the reflective journal and in-depth interview transcript below.

*"One of the main things I learned was about the awareness of the importance of sustainability aspects in the production and development of a product . . . ."* (Reflective Journal P2, 19 October 2020)

*"Sustainability is very good for human life."* (Reflective Journal P2, 19 October 2020.)

*"After carrying out this summer course program, I have a new understanding of sustainable development, especially in the agricultural sector."* (Reflective Journal P11, 19 October 2020)

*"I understand sustainability better, so I can be wiser in doing everything by minimizing things that are detrimental to the Earth."* (P2, Interviewed 22 November 2020)

The excerpts reveal that preservice chemistry teachers developed their awareness of sustainability through the program (*the importance of sustainability; new understanding*), found the value of using sustainability in life (*very good for human life*), and allowed them to be more concerned about ESD issues (*the production and development of a product*). ESD reassesses what we learn, where we learn, and how we learn. It develops the knowledge, skills, values, and attitudes that enable learners to make informed decisions and take action to solve global problems [42], such as in the agro-industry, which was a focus of this program. The program increased the preservice teachers' awareness of sustainable development by providing them with new experiences.

3.4.2. Environmental Awareness

In addition to raising sustainability awareness, the participants reported an increased awareness of their role in protecting the environment (minimizing the negative impact

on the environment) by reviewing the product production process at maximum output to measure whether the impact on the environment was maintained because the process operated as a necessity. The participants' awareness grew by having to maintain, protect, and not contribute to damage (*reducing environmental damage; more environmentally friendly; bad for the environment*) as shown in the participants' reflective journal and in-depth interview responses. A UNESCO statement [42] states that "ESD can help us understand the global nature of today's challenges, it provides us with a concrete solution for the local living environment". The principles of ESD issued by the United Nations can be used to develop an environmentally sound chemistry education that fosters and equips students with environmental attitudes, knowledge, and awareness [44]. The research project can lead to empowering communities of preservice chemistry teachers to be more concerned about environmental issues (*environmental values as well*) that impact them daily and provide a method for planning a project that addresses economic, social, and environmental values. In their reflections and interview responses, the preservice chemistry teachers stated that sustainability is an interesting concept to discuss, is important to learn, and is useful for the future. Preservice teachers participated actively in their learning to gain the information, skills, and attitudes necessary to encourage sustainable practices in their community and in their daily lives (*apply this in my daily life*). ESD can help reveal the global nature of today's challenges. It provides concrete solutions for the local environment [42]. ESD provided the preservice teachers with an experience of ESD as a practice-based approach (*bringing my own*) and encouraged them to become more concerned about ESD issues (*reducing*, *reusable*) related to their way of life.

While this research focused on the environmental awareness of prospective chemistry teachers, it is anticipated that the knowledge and understanding gained from the experience will be applied to their future teaching.

Sustainable, development-based learning, known as green chemistry, has an important role to play in chemistry education. Fibonacci at [45] states that "various chemical concepts have close links with the environment to stimulate the creativity and innovation of students to be able to use chemical concepts to solve the environmental problem". Prospective chemistry teachers can use innovative pedagogies to instill the value of sustainability in their students, as identified in the transcript below:

> *"I think it is important as a prospective chemistry teacher to study sustainability development in school because studying chemistry has something to do with the environment so it can design creativity and innovation. They can use chemical concepts to solve problems in their environment. In addition, how to implement sustainable value that can be applied to chemicals such as electrolytes and nonelectrolytes, redox, colloids, petroleum hydrocarbons, electrochemical applications, adn polymers. It can be a supporter of teacher creativity on how to analyze a characteristic of material, so that the sustainable values are embedded in students' personalities."* (P7, interviewed 22 November 2020.)

### 3.4.3. Project Development Engagement

Success in developing sustainability competencies, the learning must, of course, include supportive pedagogy. Sipos et al. at [46] stated that the most relevant pedagogy in enhancing the development of sustainable competence is a pedagogy that involves the head (cognitive domain), hands (psychomotor), and heart (affective domain). Project-based learning is a feasible model in developing sustainable competencies [43,47]. The way that project-based learning supports sustainable competency development is by encouraging students to develop sustainable competencies by providing a supportive learning atmosphere that encourages critical thinking, creative thinking, higher-order thinking skills (HoTS), problem-solving, collaboration, and communicating well.

Although the program was held online, students felt that joining the program provided them with meaningful experiences in using design thinking, discussing and solving problems, and conducting research and sharing information in discussion groups. The PjBL model offers preservice teachers a chance to experience the tangible reality of their

communities, as well as to take ownership of a project as active participants rather than mere consumers of knowledge [48].

The following excerpt demonstrates how meaningful this program was for these students:

*"It is very interesting because each group member conducts research first regarding the topic at hand. Then convey each of the ideas obtained. ... "* (Reflective Journal, P2 19 October 2020)

*"It's great for brainstorming to find solutions to existing problems. Furthermore, participants are required to carry out reliable research to construct a solution."* (Reflective Journal, P9 19 October 2020)

More detailed views were expressed during the semi-structured interviews.

*"I think critical thinking plays a very important role in finding idea for this project, such as in the process of doing research first so it doesn't necessarily give idea, besides that creative thinking is also needed, for example in determining eye-catching project name, and in my opinion problem solving is also used such as from existing problem based on previous research and how to solve the problem by giving idea in the form of a project."* (P3, interviewed 22 November 2020.)

*"Very effective learning model, because it invites the participants to think more actively. Not only analyzing but also imagining the application of the project, conducting a project, and presenting it to another member."* (P5, Interviewed 22 November 2020.)

PjBL provides a direct learning experience of diving into a problem, finding the solution, and constructing a plan to solve the problem. The prospective chemistry teacher feels fully involved in designing a project that uses agro-industry products, and they feel challenged, encouraged to think critically (critical thinking plays a very important role), brainstorm (think many things; brainstorm), conduct in-depth information searches (perform research), and think creatively through this PjBL (determining eye-catching project name).

PjBL improves the participants' capacity to address environmental and developmental issues, as well. A study conducted by [49] among preservice technology teachers in South Africa about their experience in using PjBL as a pedagogy for ESD reported that PjBL promotes social learning and a real-world context for learning about ESD. It can be used as a catalyst for raising preservice teachers' awareness of their role as agents of change.

PjBL ensures continuity with the learning experience (*to complete a group project*) in a pleasant learning atmosphere (*very exciting and interesting*), to actively develop students' critical thinking (*find* solutions), and to solve contextual problems (*to construct a solution*). This finding concurs with [40], who state that PjBL is not simply transmitted from teachers to students but is constructed in the mind of the learner as they actively engage in developing unique solutions to problems.

### 3.4.4. Communication and Collaboration

According to a body of research, collaboration and communication are key competencies in sustainability [50–52]. Students reported feeling enthusiastic about working in a multicultural environment because they could greet friends from different cultures and language groups. However, having to discuss the project compelled students to use foreign language skills while negotiating with friends from other countries. Many students stated the following:

*"The multicultural environment that occurs during the summer course is very interesting. I can communicate with people from various backgrounds. However, the biggest problem I feel is the ability to communicate in English."* (Reflective Journal P13, 19 October 2020)

*" ... ..Collaboration between members is an important aspect during project work. Even though it is virtual, because they have several group assignments, the members can be closer to one another"* (Reflective Journal, P2 19 October 2020)

*"It's very interesting because each group member conducts research related to the existing topic. Then, convey each idea obtained. Collaboration between members is an important aspect during project work."* (P8, Interviewed 22 November 2020)

The excerpts above confirm that using PjBL challenged the participants to improve their communication and collaboration skills, deepen their thinking, and gain a better understanding by performing research and presenting it to others (*conduct research related to the existing topic*). Excerpts from the reflective journals showed how the participants developed ways to communicate that improved their language skills and enabled them to effectively collaborate with their working group (*convey each idea obtained*); this ability is described as interpersonal skills to work with others and is an interpersonal competency [50,51,53,54]. Collaboration is the ability to motivate, activate, and facilitate collaboratively and participate in ongoing research and problem solving [52]. In this research,, the participants already worked collaboratively with their peers in a good and correct way, so their collaboration and communication skills were used actively.

The onset of the COVID-19 pandemic demonstrated the importance of individuals' and societies' ability to respond rapidly to unexpected risks. Collaborating to find solutions, anticipating different scenarios, negotiating trade-offs, and being ready to act quickly based on limited information are essential skills for a 21st century, global society [42]. The participants in this research project believed that collaboration is important for solving problems to produce better outcomes (an *important aspect during project work*). ESD encourages collaborative outcomes by encouraging students to work together. Collaboration means having a belief in the good of others, and, as Singh-Pillay [49] points out, it allows for individual self-introspection whilst maintaining a working relationship with people from different socioeconomic statuses. Collaboration can develop a participant's self-confidence (*I can communicate*), team-building skills (*each group member conducts research, communicate with people*), and increase tolerance related to perceptions of race, culture, and a commitment to the group.

Due to the different cultural backgrounds and participating countries, several obstacles, such as time differences, constrained the scheduling of discussions. Time management was another issue that needed to improve, as did a stronger commitment to the value of working together.

Mastering global languages, as well as an occasional unstable internet connection, caused some issues, as reported below.

*"Because during a pandemic, it is less pronounced. But still exciting. The biggest problem at hand is the timing. Because I have a group of friends from Villanova who are 11 hours apart from us, so it is challenging to match the discussion time, language is also a problem when discussing. Internet connection sometimes becomes a problem too."* (P10, Interviewed 22 November 2020)

*" . . . But the biggest problem I feel is the ability to communicate in English."* (Reflective Journal P7, 19 October 2020)

*"Compared to the language barrier, I feel more constrained by time. Classes held are sometimes too early or late at night so that it is not optimal when listening to the presentation. However, I try my best in every meeting."* (P12, Interviewed 22 November 2020)

Since the participants showed their engagement throughout this program, we can say that this summer course program encouraged them to practice and make improvements in areas such as communication and collaboration skills and obtain better knowledge about sustainability and environment awareness. Being able to collaborate well in a group and the multicultural environment also provided opportunities to link their point of view of other fields outside of chemistry. The working teams across disciplines encouraged the conceptualization of innovations and creative approaches from different perspectives towards solving the same problem [43], Ref. [55] because sustainability competencies can be implemented with scientific discipline abilities, such as social science, engineering, and business [43]. A multicultural learning environment encourages success in sustainable

preservice chemistry teacher competency development because it provides a learning atmosphere in multidisciplinary groups.

## 4. Conclusions

The findings of this study show that preservice chemistry teachers can improve their awareness of sustainability through an online ESD program. They were able to think more broadly about the idea of sustainability in many aspects of life. They become more concerned about ESD issues, which lead them to empower their communities (preservice chemistry teachers) to be more concerned about environmental issues that impact them daily. The program provided a method for planning a project that addressed economic, social, and even environmental values to slowly change their way of life. PjBL ensures continuity with the learning experience in a learning environment that actively develops students' critical thinking and the ability to solve complex contextual problems. PjBL allows students to engage and interact with diverse communities, gain more information, instill collaboration skills, and develop team-building strategies that give rise to emotional learning. These skills are critical for the future success of education to provide society with sound responses to sustainability issues. The online summer course brought new opportunities and a commitment that ESD must be included in the learning in all contexts, for all learners, regardless of their gender, location, socioeconomic status, or access to technology. In conclusion, the preservice chemistry teachers engaged in creating a project-based learning solution through an online summer course program that developed students' sustainability perspective and environment awareness, higher-order thinking skills, and communication and collaboration skills.

**Author Contributions:** Research design, data collection, data analysis, and journal writing, M.P. and N.A.P.H.; re-search design and data analysis, Y.R.; data analysis, and journal writing, E.F.; data collecting, and data analysis, J.A.S. All authors have read and agreed to the published version of the manuscript.

**Funding:** This research was funded by Universitas Negeri Jakarta, Indonesia: 3/International Research Collaboration-UNJ/LPPM/V/2020.

**Institutional Review Board Statement:** The authors' institution does not require review board statement.

**Informed Consent Statement:** Informed consent was obtained from all subjects involved in the study.

**Data Availability Statement:** Not applicable.

**Conflicts of Interest:** The authors declare there is no conflict of interest.

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
