# Peer review of "Developing Preservice Chemistry Teachers’ Engagement with Sustainability Education through an Online Project-Based Learning Summer Course Program"

_sustainability, doi:10.3390/su14031783_

Round 1

Reviewer 1 Report

I recommend that the theoretical framework be well grounded with the contributions in the field, which are quite numerous and significant and which are not reflected in the text.
The qualitative study is poor and not sufficiently clear.  The results are simple percentages that provide little information.
Much of the information that is reflected in the "Discussion" section should appear in "Results".

Author Response

No

Comments

Responses

Reviewer 1

1

I recommend that the theoretical framework be well grounded with the contributions in the field, which are quite numerous and significant and which are not reflected in the text.

We added and grounded the   theoretical framework to reflect the contributions in the field in line 36-60.

As stated “Chemistry education can play an important role in promoting sustainable development”……. What teachers think, believe and know will affect their teaching. For this reason, the development of teacher competencies is an essential key to successfully growing the value of ESD in learning.

2.

The qualitative study is poor and not sufficiently clear.

We added information of the qualitative study and the data collection in line 100-106 and 129-127.

A qualitative research method…. Multiple data collection strategies such as observations during learning activities, interviews, reflective journals, documenting assigned tasks, video, photos, and observation sheets were employed to explore the participants’ engagement. 

We completed the program information as stated “The series of activities at this stage of the process enable the participants to develop projects using agro-industry products that integrate with sustainable agro-industry business ideas….”

3.

The results are simple percentages that provide little information.

We have added more information to explain the result of our research and provide more information in line 266-269.

As stated, “The motivation for chemistry teacher candidates to join the program scored a high 84.03 %, as indicated by the following interview transcript.

“I think as a prospective chemistry teacher it is important to study SD because studying chemistry has something to do with the environment, so the students can use chemical concepts to solve problems in the surrounding environment” (P3, interviewed 22 November 2020).

The participants could see that SD is necessary and useful to study and apply in chemistry learning, which motivated them to join the program even though the material was not fully relevant to the chemistry field.

4.

Much of the information that is reflected in the "Discussion" section should appear in "Results".

We have displayed the result data in a better place and make it easier to read and understand such as move Figure 6, 7, and 8 to result part.

For example

“As shown in Figure 6, the chemical concept that students explored in this project was the antioxidant value of coffee which has the effect of reducing compounds, scavenging radical compounds, binding prooxidant metal ions and inhibiting the formation of singlet oxygen. The project demonstrates the integration of chemical concepts with economic potential that contributes to reducing environmental impacts”

Reviewer 2 Report

The topic of the article is interesting. However, the literature review and the methodology part are weak. More specifically, the issue of ESD competencies has been thoroughly studied and even in the Sustainability Journal one can find many studies such as Scherak and Rieckmann article “Developing ESD Competences in Higher Education Institutions—Staff Training at the University of Vechta”.  Also, Lozano, R.; Merrill, M.; Sammalisto, K.; Ceulemans, K.; Lozano, F. Connecting Competences and Pedagogical Approaches for Sustainable Development in Higher Education: A Literature Review and Framework Proposal. Sustainability 2017, 9, 1889. See also Makrakis, V., Kostoulas-Makrakis, N. & Kanbar, N. (2013). Developing and validating an ESD student competence framework: A Tempus RUCAS initiative. International Journal of Excellence in Education, 5 (1), 1-13. And Makrakis V, Kostoulas-Makrakis N. (2016). IN the review there is also need to discuss deeper the issue of project-based learning and in the results to show examples of such projects since the title refers to this concept and methodology. Although, it is noted that the study is qualitative, a survey questionnaire has been also used and although the results are mere descriptive there is need to discuss how the quantitative and qualitative data are used. See, the “Bridging the qualitative–quantitative divide: Experiences from conducting a mixed methods evaluation in the RUCAS programme. Evaluation and Program Planning, 54: 144-151.

Author Response

Reviewer 2

1

The topic of the article is interesting. However, the literature review and the methodology part are weak. More specifically, the issue of ESD competencies has been thoroughly studied and even in the Sustainability Journal one can find many studies such as Scherak and Rieckmann article “Developing ESD Competences in Higher Education Institutions—Staff Training at the University of Vechta”.  Also, Lozano, R.; Merrill, M.; Sammalisto, K.; Ceulemans, K.; Lozano, F. Connecting Competences and Pedagogical Approaches for Sustainable Development in Higher Education: A Literature Review and Framework Proposal. Sustainability 2017, 9, 1889. See also Makrakis, V., Kostoulas-Makrakis, N. & Kanbar, N. (2013). Developing and validating an ESD student competence framework: A Tempus RUCAS initiative. International Journal of Excellence in Education, 5 (1), 1-13. And Makrakis V, Kostoulas-Makrakis N. (2016).

We added more information to the research methodology, sustainability competencies of the participants after the program (36-60, 129-157 and 266-269).

For example

At the preliminary stage, two types of questionnaires were used to determine the participants' prior knowledge of ESD concepts and Project-Based Learning (PjBL).  The course program followed an inquiry process where a driving question challenged students to find solutions, create a product with sustainable value, receive feedback from the teacher and use it to revise their solution to present at the end of the course. The learning process began with a presentation of sustainable development material in general before focusing in on a more detailed explanation of sustainable development in the agro-industry.  Input from practitioners or entrepreneurs in the coffee, chocolate and tea industry provided students with information regarding the sustainability of the agriculture industry (agro-industry).

2.

In the review there is also a need to discuss deeper the issue of project-based learning and in the results to show examples of such projects since the title refers to this concept and methodology.

We provided a project result to show the product of PjBL in line 302-343.

For Example

The participants developed many project plans through the program, including one called COHUMA (Coffee Hush Mask). The group created an innovative face mask made from coffee husks to reduce acne-causing bacteria, disguise black spots and provide an anti-aging effect on the face. In addition to using coffee husks as the main ingredient, and to maximize the performance of the mask, the participants added vitamin E to moisturize, soften and brighten the facial skin, and to fade acne scars.  The product design can be seen in Figure 5 and the instructions for use in Figure 6).

3.

Although, it is noted that the study is qualitative, a survey questionnaire has been also used and although the results are mere descriptive there is need to discuss how the quantitative and qualitative data are used. See, the “Bridging the qualitative–quantitative divide: Experiences from conducting a mixed methods evaluation in the RUCAS programme. Evaluation and Program Planning, 54: 144-151.

We provides information of the use of questionnaire. We only use for have picture of students’ perception of ESD dan Project Based Learning to engage students in the project. We put in line 188-198.

The questionnaire used in this research was self-administered online at the beginning of the study and before providing the program material, to obtain the participants’ perceptions and interest in learning ESD-based PjBL. The instrument  was compiled based on the PjBL and ESD grids, each with its own criteria for ESD understanding, ESD and chemistry relations, PjBL in teaching chemistry, and PjBL understanding. The instrument used a Likert scale with “Very agree” to “Not agree” statements with a scale from 1 to 5.  The questionnaire results were analyzed by counting the value of the questionnaire and distributing the value into “low”, “middle”, or ”high” levels of understanding and interest in ESD and PjBL.

Reviewer 3 Report

The paper describes a summer program for pre-service chemistry teachers intended to invoke sustainability concepts via project based learning. The study has the following shortcomings which should be addressed prior to publication.

1. Number of participants (i.e. 26) is low. Please justify.
2. Questionnaires are not provided, please share in whole or in part. Also describe assessment method (was it Likert scale or subjective assessment? in case of the later, who carried out the assessment and what was the criteria for each question)
3. An outline of the summer program, nor its contents is not provided. Please elaborate.
4. Please use PjBL as acronym for Project Based Learning through out the manuscript (as done in Fig 2) to differentiate it from Problem Based Learning.

Author Response

Reviewer 3

1.

Number of participants (i.e. 26) is low. Please justify.

It is qualitative study, questionnaire used for having students’ perceptions on ESD and PjBL. We do deep observation and interview throughout the program.

2.

Questionnaires are not provided, please share in whole or in part.

We added the questionnaire in line 188-198.

Questionnaires are useful for measuring observed natural or social phenomena. The natural or social phenomenon in question is a variable in a study. The questionnaire used in this research was self-administered online at the beginning of the study and before providing the program material, to obtain the participants’ perceptions and interest in learning ESD-based PjBL. The instrument was compiled based on the PjBL and ESD grids, each with its own criteria for ESD understanding, ESD and chemistry relations, PjBL in teaching chemistry, and PjBL understanding. The instrument used a Likert scale with “Very agree” to “Not agree” statements with a scale from 1 to 5.  The questionnaire results were analyzed by counting the value of the questionnaire and distributing the value into “low”, “middle”, or ”high” levels of understanding and interest in ESD and PjBL.

3.

Also describe the assessment method (was it Likert scale or subjective assessment? in case of the later, who carried out the assessment and what was the criteria for each question)

We added information of the questionnaire in line 188-198.

The instrument used a Likert scale with “Very agree” to “Not agree” statements with a scale from 1 to 5.  The questionnaire results were analyzed by counting the value of the questionnaire and distributing the value into “low”, “middle”, or ”high” levels of understanding and interest in ESD and PjBL.

4.

An outline of the summer program, nor its contents is not provided. Please elaborate.

We added an outline of the summer program and provided the contents by Figure 1 line 127.

The research was conducted online due to covid-19 restrictions imposed in December 2019. The lessons and materials were delivered through a range of online platforms such as ZOOM, WhatsApp, Google Classroom, and YouTube. The summer course program ran from October 5th, 2020 until November 14th, 2020. The research was conducted in three stages: a preliminary stage, a research implementation stage, and a final stage. The research flow is shown in Figure 1.

5.

Please use PjBL as an acronym for Project Based Learning throughout the manuscript (as done in Fig 2) to differentiate it from Problem Based Learning.

We have revised it, thank you

Round 2

Reviewer 1 Report

The article is interesting and the subject has been covered in depth in other articles and scientific journals.  Although the authors have improved on the first proposal, I believe that there is still room for improvement in the theoretical basis, referencing recent relevant works. The methodological proposal used by the authors could be improved, especially in terms of clarity of the results leading to a more solid discussion.
I sincerely believe that it could be improved for publication.

Author Response

Reviewer 1

1

The article is interesting and the subject has been covered in depth in other articles and scientific journals.  Although the authors have improved on the first proposal. I believe that there is still room for improvement in the theoretical basis, referencing recent relevant works.

The methodological proposal used by the authors could be improved. Especially in terms of clarity of the results leading to a more solid discussion.
I sincerely believe that it could be improved for publication.

We have added more information related to referencing recent relevant work in line 34-36, 39-53, and 81-92.

As state: “ESD means creating and living a human life on Earth in a way that does not damage life but ….. “The principle of sustainable development focuses at the same time on human livings today and in the future…..  “Teachers for decades been cited as key agents in the sustainability process….”

We have added more information about the methodology and instrument line 230-233.. And we have combined the result and discussion in same part so it will be more solid and easy to understand in line 260.

For example,

To complete data collection a semi-structured interview was used to elicit students' ideas and views on ESD, the online summer course, project development, and a multicultural environment.

“This section reports the project based learning descriptions, prior knowledge of students upon the project, perceptions on online summer course, and impacts on pre-service chemistry teachers….”

Reviewer 2 Report

Despite changes in language and structure, the paper still lacks of a discussion section and adequate referencing, especially in the field of ESD. For example, the concept of ESD is mentioned 93 times in the main text and only one in the references!

Author Response

Reviewer 2

1

Despite changes in language and structure, the paper still lacks of a discussion section and adequate referencing, especially in the field of ESD. For example, the concept of ESD is mentioned 93 times in the main text and only one in the references!

For result and discussion we have combine it into same part. Hopefully it can be understand easier and clearly line 260.

As state: “This section reports the project based learning descriptions, prior knowledge of students upon the project, perceptions on online summer course, and impacts on pre-service chemistry teachers….”

We have added more information about ESD in the introduction part line 34-36 and the references already increase.

Burmeister, M., & Eilks, I. (2013). An understanding of sustainability and education for sustainable development among German student teachers and trainee teachers of chemistry. Science Education International, 24(2), 167–194. https://doi.org/10.1080/14729679.2019.1686038

UNESCO. (2005). Guidelines and Recommendations for Reorienting Teacher Education to Address Sustainability. Education for Sustainable Development in Action, 2, 44.

UNESCO. (2014). Shaping the Future We Want UN Decade of Education for Sustainable Development (2005-2014); Final Report. In United Nations Educational, Scientific and Cultural Organization. http://unesdoc.unesco.org/images/0023/002301/230171e.pdf

UNESCO. (2020). Build back better: Education must change after COVID-19 to meet the climate crisis. In 18.06.2020. https://en.unesco.org/news/build-back-better-education-must-change-after-covid-19-meet-climate-crisis

Reviewer 3 Report

The authors have made adequate changes in the revised manuscript for publication.

Author Response

Reviewer 3

The authors have made adequate changes in the revised manuscript for publication.

Thank you for your comments.

Round 3

Reviewer 2 Report

You attempted to upgrade the paper in line with the comments.